# Composable Semi-parametric Modelling for Long-range Motion Generation

## Abstract

Learning diverse and natural behaviors is one of the longstanding goal for creating intelligent characters in the animated world. In this paper, we propose "COmposable Semi-parametric MOdelling" (COSMO), a method for generating long range diverse and distinctive behaviors to achieve a specific goal location. Our proposed method learns to model the motion of human by combining the complementary strengths of both non-parametric techniques and parametric ones. Given the starting and ending state, a memory bank is used to retrieve motion references that are provided as source material to a deep network. The synthesis is performed by the deep network that controls the style of the provided motion material and modifies it to become natural. On skeleton datasets with diverse motion, we show that the proposed method outperforms existing parametric and non-parametric baselines. We also demonstrate the generated sequences are useful as subgoals for actual physical execution in the animated world. Please refer to our project page [1] for more synthesised results.

## 1 Introduction

When faced with a specific goal in another location, humans can effortlessly find multiple distinctive trajectories and control their body to approach the goal with diverse and natural behaviours. However, we are still at the early stage for such sophisticated controlling of simulated characters. Recent reinforcement learning approaches (Peng et al., 2018) struggle to generate diverse motion. Other methods like imitation learning (Ye & Alterovitz, 2017; Aleotti & Caselli, 2006; Lawitzky et al., 2012) generalize badly to large scale demonstration data. The heart of this challenge lies in that real-world human behaviours are inherently multi-modal distributed. Direct behaviour learning is difficult without access to the explicit distribution of motion states.

In this paper, we take a step toward generating long-range, diverse and physically plausible motion sequences given starting and ending states, rather than learning a policy for a physical simulator. Meanwhile, we expect the model could generate novel behaviour, i.e., unseen motion in training set. This has several valuable applications: (1) Synthesised vivid motion for animation production without excessive human labor. (2) Generated novel behaviours for player customization of action skills in video games. (3) Interpolated sequences as subgoals for policy training through reinforcement learning (Peng et al., 2018).

To fulfil the above requirements, we need to address the following difficulties: (1) How to guarantee both diversity and naturalness, which is usually trade-off in the domain of generation (Srivastava et al., 2015)? (2) How to achieve long-range behaviour synthesis, which is stuck by error accumulation problem in many temporal modelling tasks (Denton & Fergus, 2018)? (3) How to generate unseen behaviours without loss of diversity or naturalness, which is hardly addressed in previous researches? As shown in Fig 1, the current two main branches of motion generation methods, i.e., parametric and non-parametric, are yet to deal with these difficulties properly. Parametric (Goodfellow et al., 2014) (e.g., GAN) methods could not maintain the reality/naturalness of generated sequences, and diversity (e.g., VAE) is hard to preserve. Non-parametric methods (Haarbach et al., 2018) involving motion clip copy/paste or blending has less superiority in smooth transition based on diverse reference sequences.

---

[1]https://sites.google.com/view/cosmo-supp/home

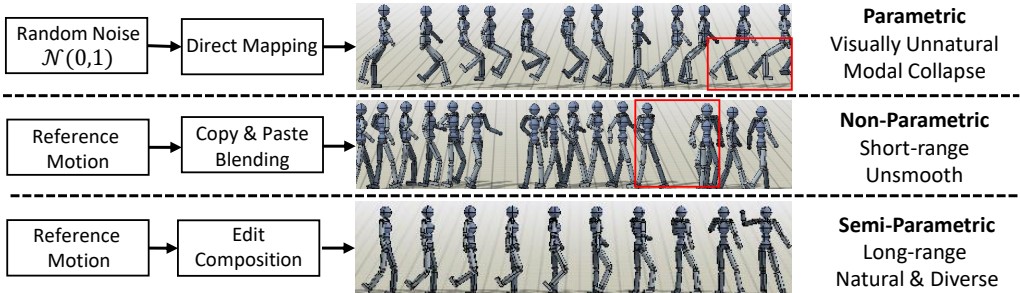

Figure 1: Comparison of different motion generation schemes. Three rows from top to bottom correspond to generated results of parametric, non-parametric and semi-parametric modelling respectively. Parametric model like GAN is prone to generate unnatural motion (twisted body in top row). Meanwhile the modal collapse is another longstanding issue for parametric model. Non-parametric model like simply copy and paste is infeasible to get visually smooth and long-range motion (second row). On the contrary, Our proposed semi-parametric model i.e., COSMO, is able to generate long-range motion sequences with diverse behaviours.

In this work, we propose COmposable Semi-parametric MOdelling (COSMO), which is a method that can leverage the large spectrum of motion skills from unlabeled data in a semi-parametric manner. Our method combines the complementary strengths of both non-parametric techniques and parametric ones. First, to avoid mode collapse in the sequences, we initially sample reference subsequences from a held-out reference set to encourage multi-modal behaviors in the generated sequence, which is mainly inspired from non-parametric methods (Haarbach et al., 2018). Second, we propose a self-supervised disentanglement model for extracting the *content* and *style* from each reference subsequence respectively. Here *content* refers to the characteristic of state at each time step, e.g., moving speed, direction, and general gesture of upper body, while *style* refers to the long-range motion pattern which keeps relative constant across whole subsequence. An embedded latent space is constructed where the learned style vectors could be combined to obtain novel style feature. A new subsequence is obtained by composing content and style freely from different reference subsequences. Considering that this operation only produces a single subsequence, we refer this step as **local motion composition**. Finally, for all new sequences, we compose them along the temporal direction in a sequential order. However, from a hodgepodge of reference segments, it is non-trivial to generate natural and meaningful behaviors. To guarantee naturalness at long-range scale, we then propose to use goal conditioned bi-directional interpolation for modeling the long-range nature needed in the task. This step is regarded as **global motion composition**, which covers the whole temporal scale (often more than 100 steps). On two human motion datasets, we show that the proposed method outperforms existing parametric and non-parametric baselines. We also demonstrate the generated sequences are useful as subgoals for actual physical execution in the animated world.

Our paper makes three contributions. First, we proposed COmposable Semi-parametric MOtion generation (COSMO), a method that can generate the required sequences in a semi-parametric, composed way. Second, we conducted experiments on CMU Mocap dataset [2] and SFU Mocap dataset [3] and shows our method outperforms strong baselines such as VAEs and GANs significantly. Third, we proved empirically that the generated sequences can serve as subgoals to learn an actual policy.

## 2  METHOD

### 2.1  SEMI-PARAMETRIC MODELLING FOR MOTION GENERATION

Given starting and ending states (denoted as $\mathbf{s}_0$ and $\mathbf{s}_L$ respectively) as inputs, our goal is to synthesize intermediate states that have natural transition and diverse behaviors. As illustrated in Fig 2, $N_R$ reference subsequences (denoted as $\mathbf{R}^i, i = 1, .., N_R$) are sampled and fed into local motion composition

---

[2]http://mocap.cs.cmu.edu/
[3]http://mocap.cs.sfu.ca/

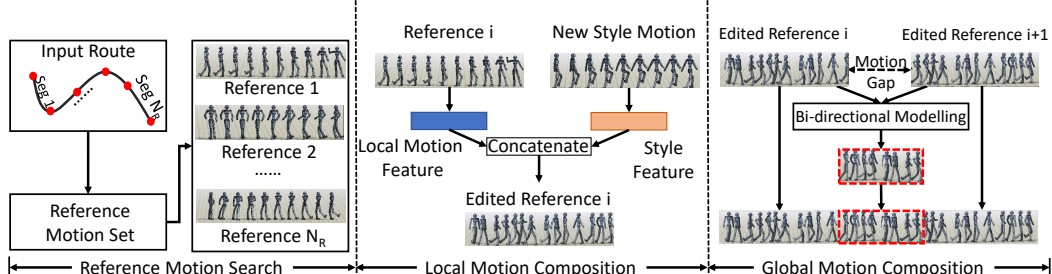

Figure 2: Proposed framework for motion generation. Step 1: We search for reference subsequences within a held-out set. Step 2: A novel subsequence is then generated based on reference one with local motion composition. Step 3: All edited subsequences are connected together in temporal order with global motion composition.

sub-module $\phi_{loc}$. Invalid subsequences are first filtered out with predefined requirements and please refer to appendix 6 for detailed description. $\phi_{loc}$ is used for altering the motion style of reference subsequence $\mathbf{R}^i$ based on another sequence $\mathbf{R}$ as input or directly sampling a style feature from learned latent space: $\hat{\mathbf{R}}^i = \phi_{loc}(\mathbf{R}^i, \mathbf{R})$, where $\mathbf{R}$ is optional input. With this in hand, we compose edited subsequences $\{\hat{\mathbf{R}}^i\}_{i=1}^{N_R}$ along temporal direction smoothly. Concretely, we generate a new clip $\tilde{\mathbf{R}}^{i-1:i} = \phi_{glo}(\hat{\mathbf{R}}^{i-1}, \hat{\mathbf{R}}^i)$ which bridges the gap between $\hat{\mathbf{R}}^{i-1}$ and $\hat{\mathbf{R}}^i$. The final composed sequence is obtained through concatenation along the temporal direction for all subsequences and generated clips.

**Why semi-parametric modelling?** Regarding both motion category and transition dynamics (from current to next state), human behaviors are intrinsically multi-modal distribution (Holden et al., 2017). Parametric models (e.g., VAE (Kingma & Welling, 2014) or GAN (Goodfellow et al., 2014)) assume prior distribution to be uni-modal and predefined, which is not suitable to handle real world motion sequences (Kingma & Welling, 2014; Arora & Zhang, 2017). Non-parametric model does not require explicit prior distribution but not able to deal with long-range and highly diverse motion properly, mainly restricted by modelling capability. We also conduct targeted comparison experiments in Section 4. Compared to parametric methods, semi-parametric modelling naturally guarantees motion diversity by introducing reference sequence also maintaining a large degree of freedom for generation. Moreover, semi-parametric modelling does not require access to or a model of data distribution. It implies that our method is also directly applicable for unseen motion sequence.

Constrained by starting and ending states with unchangeable locations, it is not feasible to randomly sample reference subsequences for generation. To construct a complete motion sequence, we execute a manually designed procedure for searching reference subsequences. For detailed description, please refer to appendix 6

## 2.2 Local Motion Composition with Content/Style Disentanglement

**Local motion composition stands for generating a relative short-range of motion clip based on reference subsequences as inputs.** Content and style features are learned jointly, whose free-form composition is used for synthesising new motion sequence. We first describe the motion style modelling part.

**Motion Style Modelling.** Style is considered as motion information which stays the same throughout the whole subsequence. To this end, we extract style pattern by conducting feature fusion along the temporal direction. More specifically, for a sequence with $T$ frames, we gradually fuse all frames into one constant feature with $C_s$ channels (denoted as $\mathbf{h}_s$). We adopt $1D$ convolution along the temporal direction for each operation. Three convolution layers are stacked together with a kernel size of $(T/2, T/4, T/8)$ respectively, where large kernel size facilitates capturing motion style throughout the whole subsequence.

To get better generalization ability for unseen reference or starting/ending states, we propose to construct an embedding space for style pattern modelling. More specifically, a set of embedding vectors (denoted as $\mathbf{H}_s = \{\mathbf{h}_s^i\}_{i=1}^M$) are learned together with above convolution layers (denoted as $\psi_{sty}$). The corresponding outputs (denoted as $\mathcal{A}_s \in R^M$) of $\psi_{sty}$ are a family of coefficients for a weighted sum of $\mathbf{H}_s$. This is mainly inspired by the intuition that complex human activity could be considered as a combination of several simple actions. When trained with sufficiently large motion data, the model tends to learn basic style patterns for human motion. Compared to directly outputting a feature vector, this helps better generalize to unseen motion during testing. The final style feature (denoted as $\mathbf{f}_s \in R^{C_s}$) is generated as follows:

$$\mathbf{f}_s^i = \psi_{sty}(\mathbf{R}^i)\mathbf{H}_s. \tag{2.1}$$

**Motion Content Modelling.** In this part, we focus on motion dynamics reflected by each single state. To this end, 1D convolution layer with kernel size of 3, i.e., covering consecutive 3 states, is adopted for content modelling. The output channel of this layer is designed less than original motion dimension for extracting the most relevant information. This is empirically set to 5 as a design choice. Meanwhile, velocity, position for root joint and step pattern for foot joint are fed as inputs. We denote all the above factors as $\mathbf{f}_c$ with $T$ steps and $C_c$ channels. During local motion composition, style feature $\mathbf{f}_s$ is combined with content feature $\mathbf{f}_c$ through concatenating along the channel axis:

$$\mathbf{f}_c^i = \psi_{cnt}(\mathbf{R}^i), \hat{\mathbf{R}}^i = \psi_{rec}(\mathbf{f}_c^i, \mathbf{f}_s^i), \tag{2.2}$$

where $\mathbf{f}_s$ is tiled $T$ times to match with $\mathbf{f}_c$ and $\psi_{rec}$ outputs reconstructed $\hat{\mathbf{R}}^i$. $\psi_{rec}$ follows stacked three layer 1D convolution operation with kernel size of 3. Finally, reconstruction loss is used to learn the style and content feature jointly as follows:

$$\mathcal{L}_{rec}^R = \frac{1}{N_R}\Sigma_{i=1}^{N_R}||\hat{\mathbf{R}}^i - \mathbf{R}^i||_2^2. \tag{2.3}$$

## 2.3 Global Motion Composition via Goal Conditioned Bi-directional Modelling

**Global motion composition is connecting edited short-range subsequences into a completed and long-range one.** Generating smooth and natural transition between $\hat{\mathbf{R}}^i$ and $\hat{\mathbf{R}}^{i+1}$ is critical to obtain a long-range sequence. The major challenge comes from large states variation between $\hat{\mathbf{R}}^i$ and $\hat{\mathbf{R}}^{i+1}$. Inspired by recent work of Bi-LSTM modelling (Ma & Hovy, 2016), bi-directional motion composition is utilized for this part. More specifically, motion states are first mapped to a higher dimensional space with $\varphi_{enc}$ and $\varphi_{dec}$, which is defined as follows:

$$\mathbf{f}_t^i = \varphi_{enc}(\mathbf{r}_t^i), \hat{\mathbf{r}}_t^i = \varphi_{dec}(\mathbf{f}_t^i). \tag{2.4}$$

Both $\varphi_{enc}$ and $\varphi_{dec}$ are 1D convolution layers. Then we predict the possible states at both directions with $\mathbf{f}_t^i$, where higher dimension (256) provides a sparser space enabling better composition results. For forward prediction, we take $\mathbf{f}_{T-4:T}^i$ as inputs to obtain forward states $\mathbf{f}_1^{i:i+1}$. Meanwhile, $\mathbf{f}_{1:5}^{i+1}$ are fed as inputs treated as goal conditions. We conduct this procedure in a recurrent manner with $P$-step prediction in total. Predicted $\mathbf{f}_1^{i:i+1}$ is concatenated with last four states forming next step inputs. Backward direction takes exactly the reverse procedure, i.e., $\mathbf{f}_{T-4:T}^{i+1}$ are treated as inputs while $\mathbf{f}_{1:5}^i$ are as goal:

$$\mathbf{f}_{p+1}^{i:i+1} = \varphi_{fpre}(\mathbf{f}_{p-4:p}^i, \mathbf{f}_{1:5}^{i+1}), \mathbf{f}_{p+1}^{i+1:i} = \varphi_{bpre}(\mathbf{f}_{p-4:p}^{i+1}, \mathbf{f}_{T-4:T}^i), \tag{2.5}$$

where $\varphi_{fpre}$ and $\varphi_{bpre}$ share the same architecture.. Moreover, we utilize another model to produce coefficients for the weighted sum of outputs. In this way, we are able to construct more flexible latent space to facilitate composition:

$$c = \varphi_{com}(\mathbf{f}_p^{i:i+1}, \mathbf{f}_{P-p}^{i+1:i}, p), \hat{\mathbf{f}}_p = c\mathbf{f}_p^{i:i+1} + (1-c)\mathbf{f}_{P-p}^{i+1:i}. \tag{2.6}$$

$\varphi_{com}$ is one layer 1D convolution with sigmoid function for producing $c$. More specifically we introduce $p$ and linearly map it to $\hat{p} \in (-1, 1)$ and $c = \hat{p} + 0.1 * \sigma(conv([\mathbf{f}_p^{i:i+1}, \mathbf{f}_{P-p}^{i+1:i}]))$. During training, we randomly select a motion clip with a length of $10 + P$. First as well as last 5 states are fed as inputs to get intermediate $P$ states. Reconstruction loss is used for training:

$$\mathcal{L}_{rec}^f = \frac{1}{P}\Sigma_{t=1}^P||\hat{\mathbf{r}}_t - \varphi_{dec}(\hat{\mathbf{f}}^t)||_2^2. \tag{2.7}$$

Compared with goal-conditioned single forward prediction model, the starting and ending states are treated equally in our model, which avoids drifting away from the ending state. Meanwhile, we provide corresponding empirical results in Section 4. After global motion composition, final composed sequence is directly obtained by concatenating all edited subsequences (local part) as well as transition clips (global part) along the temporal direction.

We use tensorflow (Abadi et al., 2015) to implement all our models. All activation function used in our model is ReLU operation. During training, learning rate is set to 1e-4 and optimized with Adam optimizer (Kingma & Ba, 2015). $\beta_1$ and $\beta_2$ are set as 0.9 and 0.999 respectively. All models are trained with 30 epochs in total. Note that the local and global motion composition submodule do not need joint training. They are used jointly during testing.

## 3 RELATED WORK

**Motion Interpolation.** Given start and end states, this task aims to synthesize intermediate states which smoothly translate between them (Urtasun et al., 2008). For video interpolation (Liu et al., 2017; Li et al., 2019; Meyer et al., 2015; Niklaus et al., 2017) where start and end states are two consecutive frames, the final result is expected to increase frame rate of original video to a higher value. Previous researches often utilize phase dynamics (Meyer et al., 2015), flow based feature (Liu et al., 2017) and other motion information (Niklaus et al., 2017) to facilitate this task. Our work is different from this branch of work because there exists large motion gap between start and end states in our settings. Another branch of work is video completion (Cai et al., 2018; Li et al., 2019; Wexler et al., 2007). It receives two *nonconsecutive* frames as input and aims to fill the motion gap between start and end states. Cai et al. (2018) firstly attempts to solve this task and more specifically, propose to select out a rational path in the latent space with BFGS (Byrd et al., 1995) algorithm. Li et al. (2019) incorporates the 3D convolution layers and LSTM network into a unified model, which tries to automatically find the optimal results for intermediate frames. Despite much progress has been made in this filed, the high dimensional data (i.e., video frames) severely restricts video completion within *simple* and *seen* motion categories. However, we do not limit the start and end states belonging to the same sequence. Meanwhile, we expect the interpolated sequence as diverse as possible meanwhile with natural transition between synthesised states. This has not been deeply addressed in previous motion completion works (Xia et al., 2019; Lee et al., 2019). As a potential downstream application, our model could be used to construct motion planning (Myers, 1983) algorithm. Compared to goal-driven RL (Kulkarni et al., 2016; Lee et al., 2019), our model gets rid of requirements hard to achieve, i.e., known dynamics of agent, which is more general and applicable to more motion planning scenarios.

**Temporal Data Prediction.** There have been many researches focusing on temporal data prediction. This task aims to infer the future possible states conditioned on history states as input. Video prediction (Srivastava et al., 2015) takes a major part in this field, which involves pixel-wise forecasting for every following frames. Previous methods (Denton & Fergus, 2018; Babaeizadeh et al., 2018; Denton & Birodkar, 2017; Finn et al., 2016) have made it to produce high quality prediction on bouncing digits (Srivastava et al., 2015), robot motion (Finn et al., 2016) and semantic map (Jin et al., 2017). However, recent work (Xu et al., 2018a) still encounters much difficulties in forecasting complex movements involving articulated subject (Wichers et al., 2018). Moreover, these works are prone to have motion blur (Finn et al., 2016) and error accumulation (Denton & Fergus, 2018) problems. Out of above concerns it is more appropriate for us to formulate our method as interpolation instead of prediction model. In terms of lower dimension data, human pose (Sun et al., 2019; Zhang et al., 2019) as well as path trajectory (Xu et al., 2018b) are also hot spots in this task. Several works adopt probabilistic Bayesian model (Bhattacharyya et al., 2019) to dig out latent factors which influence future dynamics. Another branch of researches (Gupta et al., 2018; Xu et al., 2018b) utilize deep recursive model (e.g., LSTM Hochreiter & Schmidhuber (1997)) to extract critical feature for prediction. Low dimension data enables all these works could easily scale to multi-target settings (Zhao et al., 2019). However, prediction model loses its power when encountering out of distribution data (e.g., unseen motion category in test set). As comparison, our interpolation model could naturally generalize to unseen motion facilitated by semi-parametric modelling, which gets rid of explicit representation for future dynamics.

**Generative Semi-parametric Modelling.** For generative task such as image synthesis (Qi et al., 2018), translation (Wang et al., 2019) and inpainting (Iskakov, 2018), semi-parametric modelling achieves considerably more realistic visual quality and better style consistency between source and target images. Semi-parametric modelling is also utilized in reinforcement learning (Kulkarni et al., 2016) for navigation task (Eysenbach et al.). More specifically, it searches rational path on replay buffer, which enables agents to solve sparse reward tasks over one hundred steps. Wang et al. (2017) combines semi-parametric modelling and VAE for large-scaled motion affordance. In this work we use semi-parametric modelling for two main reasons: (1) it could substantially increase the length of interpolated sequence through acting as example guidance, (2) it enables our model to synthesise visually appealing results even with out of distribution inputs. Meanwhile, we would like to highlight that to our best knowledge this is the first attempt to use semi-parametric modelling on motion interpolation task.

**Motion Generation in Computer Graphics.** In the context of computer graphics, there is a branch of researches (Kovar & Gleicher, 2003; Park et al., 2002; Levine et al., 2012; Tan & Tai, 2012; Holden et al., 2016) which also concentrate on motion generation, i.e., obtaining a continuous trajectory from a discrete set of poses. Haarbach et al. (2018) provides a comprehensive study on this topic. It analyzes the characteristics of higher order rigid body motion interpolation methods. Our works shares similar target with this branch of work. However, we would like to emphasise that these works (Tan & Tai, 2012; Kovar & Gleicher, 2003; Park et al., 2002) are in parallel with ours and have completely different research routine on this task. More specifically, graphics methods focus on finding a optimal and explicit mathematical solution regardless of input motion sequences. Different from them, our work is data-driven and encourages both reality and diversity of interpolated results.

**Imitation Learning.** Imitation learning (Pomerleau, 1989; Ye & Alterovitz, 2017; Lawitzky et al., 2012) is commonly adopted as a standard method in the domain of robotics and many other areas. Behavioral Cloning (Osa et al., 2018) is one of the underlying approaches that utilize a demonstrations as supervisory signal. The most relevant work to ours is (Peng et al., 2018) which leverages reinforcement learning to imitate natural motions. However, our method does not imitate specific motion trajectories but to generate natural and diverse actions that are reasonable between given states. Our method can be further differentiated from past literature in two aspects: (1) our method is orthogonal to imitation learning because it interpolates states rather than predicts actions when states are given. More importantly, (2) the proposed method is able to produce between *unseen* states while imitation learning focuses on imitating and capturing the demonstrated distribution.

## 4 EXPERIMENTS

### 4.1 EVALUATION SETTINGS

We use CMU Mocap [4] and SFU Mocap [5] datasets for evaluation. Both datasets contain diverse daily human motion sequences which are suitable for training. Considering that the original sequences possess different length, we get reference subsequence in a sliding window manner, where $T = 120$ and $P = 40$. For both datasets the dimension of state is 63, which is 21 joints with 3D coordinates. Here we would like to emphasise that we keep a held-out reference set (denoted as $\mathcal{D}_R$) from training data (denoted as $\mathcal{D}_T$) for further testing, to demonstrate that our model could generate novel behaviour never seen during training. More specifically, $\mathcal{D}_R$ is used for sampling novel motion and starting/ending states during testing. Note that the following all experiments are with SFU Mocap datasets. For the visual results on CMU Mocap datasets, please refer to our project page.

For local motion composition, we compare our model with two strong parametric baselines: VAE (Kingma & Welling, 2014) and GAN (Goodfellow et al., 2014). For fair comparison, we train both models with all data, i.e., $\mathcal{D}_T$ and $\mathcal{D}_R$. Meanwhile we follow Yan et al. (2018) and Barsoum et al. (2018) about the hyper-parameter setting of both models. But the input dimension is adjusted to match our data. For global motion composition, we compare our model with temporal prediction baselines: (1) **Baseline1**: $\varphi_{fpre}$ without last 5 states as goal condition, (2) **Baseline2**: $\varphi_{fpre}$ with last 5 states as goal condition, (3) **Baseline3**: both $\varphi_{fpre}$ and $\varphi_{bpre}$ but without last/first

---

[4]http://mocap.cs.cmu.edu/

[5]http://mocap.cs.cfu.ca/

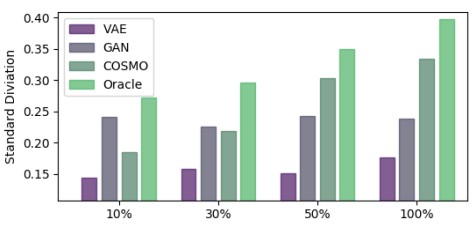
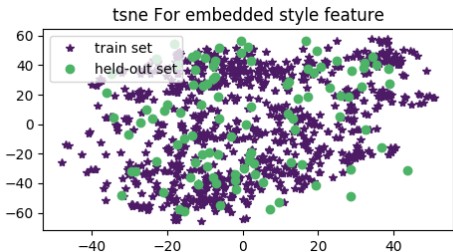

Figure 3: Motion diversity evaluation.    Figure 4: Style feature visualization

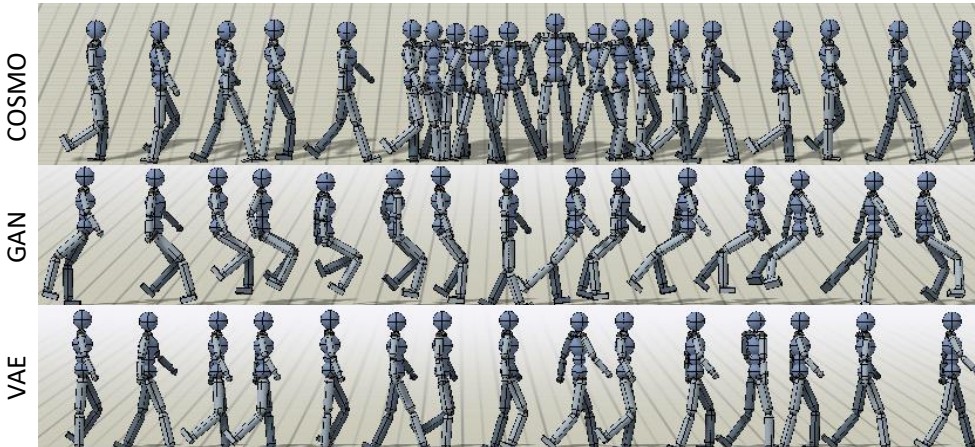

Figure 5: Comparison of motion naturalness with VAE and GAN based models. Best view in color.

5 states as goal condition. All three baselines are trained with the same hyper-parameter setting as our global composition model.

## 4.2 EVALUATION FOR LOCAL MOTION COMPOSITION

**Is generated motion diverse enough?**  We compare our model with parametric model (i.e., VAE (Kingma & Welling, 2014) and GAN (Goodfellow et al., 2014)) in evaluation of motion diversity. For the majority of generative models, the diversity of generated result is upper bounded by training data Barratt & Sharma (2018). Motivated by this, we compare motion diversity with VAE (Kingma & Welling, 2014) and GAN (Goodfellow et al., 2014) based models under different percentages of data used for training. As illustrated in Fig 3, 10%, 30%, 50% and 100% training data are used respectively. After training converged, we calculate the averaged standard diviation of all joints with a higher value indicating more diverse. We can see that both parametric models keep relative constant motion diversity which is comparable with training data used less than 30%, but largely inferior to the diversity of full training data. On the contrary, with an increasing percentage of data used for training, our model achieves higher motion diversity, which mainly benefits from the general semi-parametric modelling framework.

**Is generated motion visually natural?** For evaluation of visual naturalness, we provide generated motion results and compare with VAE (Kingma & Welling, 2014) and GAN (Goodfellow et al., 2014). As shown Fig 5, from top to bottom generated motion sequences correspond to our model (COSMO), GAN (Goodfellow et al., 2014) and VAE (Kingma & Welling, 2014), respectively. Note that all three sequences are with a length of 440 time steps. Our model (COSMO) generate the sequence with three reference subsequences. While both GAN (Goodfellow et al., 2014) and VAE (Kingma & Welling, 2014) models directly produce the motion in a recurrent manner. The GAN based model fails to synthesise a normal walking sequence with large pose distortion. The VAE based model is able to generate a visually natural walking sequence facilitated by KL loss during training but struggles to produce diverse motion behaviour. Different from all these parametric models, Fig 5

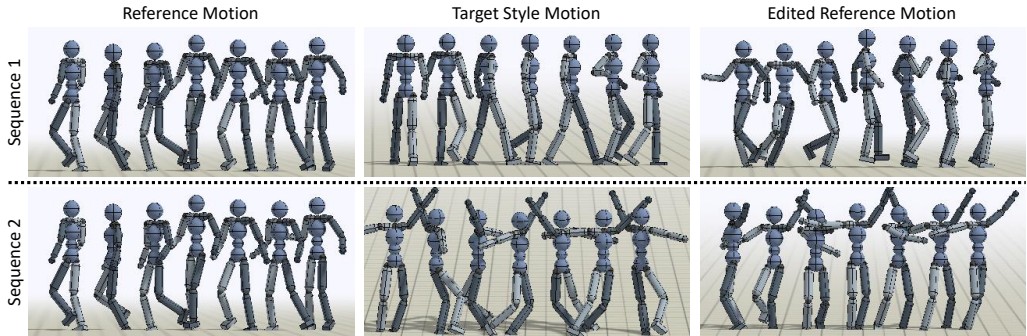

Figure 6: Visualization of local composed reference sequence.

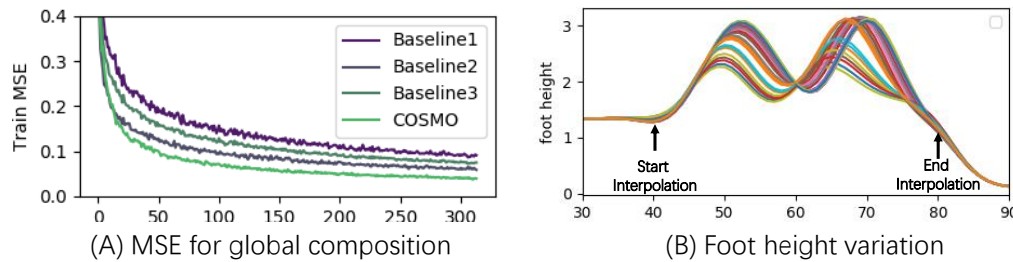

Figure 7: Evaluation of global composition in terms of training MSE and foot height variation.

shows that our semi-parametric modelling based model achieves natural state transition throughout the whole sequence, meanwhile provides natural and diverse motion (i.e., walking-turning-walking) behaviours during generation. **We encourage readers to refer to our project page for more video demonstrations**.

**Do proposed model learn a meaningful style space?** Recall that for representation learning of style feature, we construct an embedded space instead of directly generating style feature. In this way, we expect our model able to map a novel style pattern into learned embedded space. For validation we randomly select $10\%$ training data then extract corresponding style feature (denoted as $\mathbf{S}_{train}$) with $\varphi_{sty}$. Regarding the held-out set (denoted as $\mathbf{S}_{ref}$) we conduct the same operation for all reference subsequences. We visualize the distribution of both $\mathbf{S}_{trn}$ and $\mathbf{S}_{ref}$ with t-SNE (van der Maaten & Hinton, 2008) in a two-dimensional plane. As shown in Fig 4, the purple dots indicate training data while green dots stand for reference data. We can see that the style feature of training data spread evenly across the plane. Meanwhile, the majority of style of feature of reference subsequences is covered by that of training data. Within the learned style space, embedded layer acts as a set of *style* bases where novel style can be approximated by the combination of these bases.

**Can COSMO compose two reference subsequences into a novel one?** Part of the generation diversity of our model results from free composition of reference subsequences. As shown in Fig 6, we provide two edited sequence examples which possess the general motion style from one subsequence, but detailed motion pattern from another one. Taking the second sequence (bottom row in Fig 6) for example, the target style motion shows a spinning motion with both hands raised up (style), while the reference motion is a regular walking sequence. We can notice that the final edited reference motion (third column, bottom row) fully captures the style of upper body meanwhile maintains the walking pattern from reference sequence. Moreover, both top and bottom rows use the same reference motion, but with different style as inputs, our model can still produce highly diverse behaviours. For the space restriction of paper writing, we provide more examples whose new style features are sampled from constructed embedded space. We encourage readers to refer to our project page for more video demonstrations.

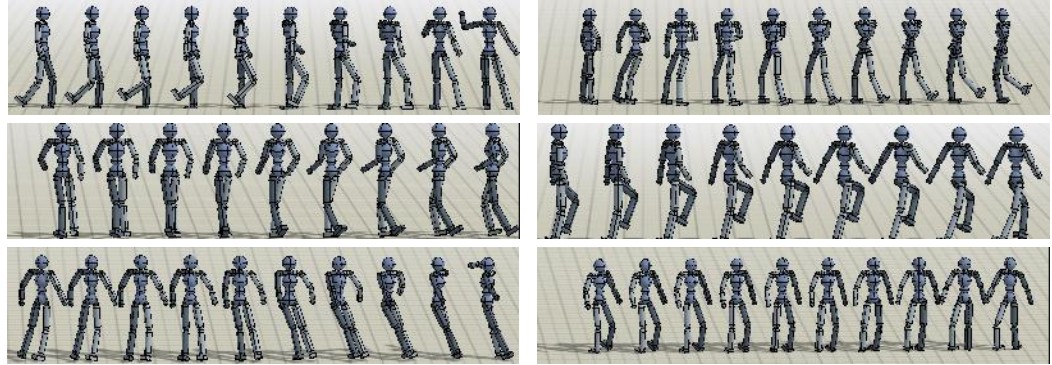

Figure 8: States transition visualization for evaluation of smoothness.

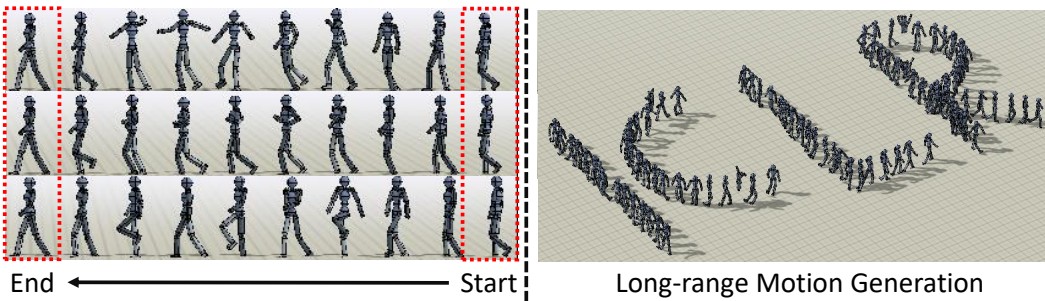

End ←——————————————————— Start          Long-range Motion Generation

Figure 9: Left: Motion generation given the same starting and ending states. Right: Long-range Motion generation with characteristic routes.

## 4.3 EVALUATION FOR GLOBAL MOTION COMPOSITION

**Do COSMO learn better motion transition than basic prediction models?** Different from prediction model, COSMO produces motion transition given starting or ending states from different sequences, i.e., there is no ground truth for evaluation. To this end, we provide training error for evaluation of motion transition. As shown in Fig 7(A), we can see that our model (COSMO) outperforms other methods by a large margin. Base1 achieves lowest accuracy caused by no ending states are provided during composition. Baseline2 and Baseline3 models perform better than base1 model, which indicates that both bi-directional and goal-conditional modelling scheme facilitates motion composition by a large margin. Our global composition sub-module combines the strength of both Baseline2 and Baseline3 models, which evenly utilize the information from starting and ending states. However, merely considering the training error is not sufficient. Next, we further evaluate the performance of our model given two different sequences.

**Can COSMO guarantee smooth transition between two different sequences?** Fig 8 demonstrates motion transition results given starting and ending states from different reference subsequences, respectively. Note that for all sequences shown in Fig 8, the starting as well as ending states are from held-out reference set. We can observe that our model is able to generate smooth and natural transition when starting and ending states are similar. Moreover, when encountered large motion difference, e.g., from walking to greeting, turning back with a relatively large degree, our model still makes it to generate visually natural transition sequence. **We encourage readers to refer to our project page for more video demonstrations**.

**Is global motion composition merely linear interpolation?** One possible trivial solution learned by our transition model is to simply linear interpolation between starting and ending states. However, common human motion generally involves non-linear trajectories. Linear interpolation is prone to produce unnatural motion which is easy to detect by human eyes. To valid that whether our model conducts linear interpolation between two sequences, we record the height variation of the right foot

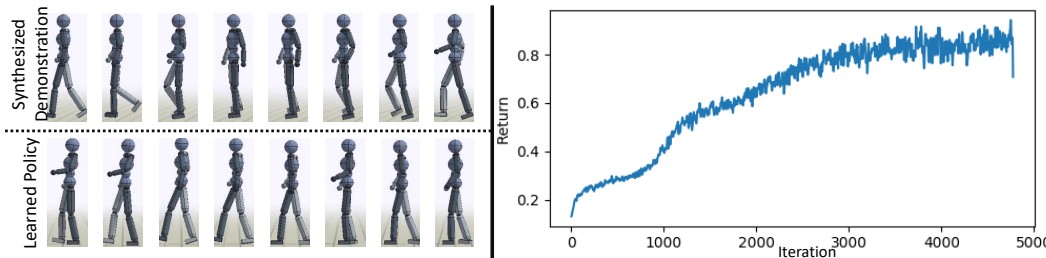

Figure 10: Generated results as demonstration guidance for imitation learning.

in a composed motion sequence. Meanwhile, we manually rotate the second sequence w.r.t. the final state of the first sequence to show that our model is robust to a large range of direction difference between two sequences. As shown in Fig 7(A), two black arrows indicate the starting and ending steps for composition. Here we present multiple curves which correspond to different rotation angle mentioned earlier. All recorded curves are highly non-linear but smooth between starting and ending points. Moreover, our model adaptively changes foot height with different rotation configurations, which indicates smooth and natural motion for motion composition.

**Visualization of final composed sequences.** Combining local and global motion together, we are able to generate final sequences. Recall that our model is constrained by given starting and ending states based on motion interpolation. To this end, we present three composed sequences with a length of 480 time steps, i.e., three edited reference sequences (length of 120) and four generated clips for global motion composition. As shown in left part of Fig 9, starting from the same state, we are able to generate long-range and visually natural motion boosted by the local and global motion composition. Meanwhile, we are able to generate diverse behaviour (shown as complex hand and foot motion) facilitated semi-parametric modelling. **We encourage readers to refer to our project page for more video demonstrations**.

### 4.4 APPLICATION

In this section, we present several downstream applications related to our motion generation model. The first one is diverse motion generation under fixed rout constrain. The second one is demonstration motion guidance for imitation learning (Peng et al., 2018).

**Diverse motion generation.** As shown in the right part of Fig 9, we manually design four routes for motion generation. We can see that our semi-parametric model is able to produce intermediate motion states clearly following the predefined route. Note that the longest motion sequence indicating character "R" is over 1500 time steps. **We encourage readers to refer to our project page for corresponding video demonstrations**.

**Expert demonstration guidance for imitation learning.** Under a simulated environment with gravity constrain (Peng et al., 2018), unnatural motion violating physical law (e.g., severe joint twisting) is hard for a simulated agent to follow. To further show our model produces realistic motion, generated results are used for demonstration guidance of imitation learning. As shown in Fig 10, the left part is demonstration synthesised by COSMO (top) while the bottom one is learned policy with Peng et al. (2018). We can see that the learned motion succeeds to follow the synthesised one. The right part is return curve during training, which also shows that our generated motion is realistic enough for the guidance of imitation learning. **We encourage readers to refer to our project page for corresponding video demonstrations**.

### 5 CONCLUSION

In this work, we propose to generate long-range motion in a semi-parametric way. We first sample reference motion subsequence from the held-out set and change the motion style with a local motion composition scheme. We then compose all reference subsequences with the proposed global motion composition scheme. Given the same starting and ending states, the proposed model is able to

generate long-range, diverse and natural motion sequences over 1000 time steps without loss of visual quality.

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

## 6    APPENDIX

**Requirements for reference sequence $\mathbf{R}^i$.** Our model does not restrict the length of $\mathbf{R}^i$, neither any specific/predefined starting or ending states of it. However, there are still restrictions for $\mathbf{R}^i$:

- **Physically reachable**. In this work we do not have access to the dynamics of other objects, which should be excluded from candidate references if involved. For example, our model does not intend for synthesising transition sequence from running to lifting. The latter behaviour needs dynamic information (bell) to model. Therefore, in our experiments reference subsequence is first restricted to single character motion without external interaction.

- **Moving on flat ground**. Besides physically reachable, we also restrict the character moving on flat ground. Because in our task there is no predefined ground information available for modelling. To get rid of unexpected composition results, we filter out these sequences recorded from uneven terrain.

**Searching procedure for reference subsequences**

- Step 0: Calculate overall root locations for all reference subsequences. We denote corresponding minimum/maximum value as $\mathbf{t}_{min}/\mathbf{t}_{max}$ respectively.

- Step 1: Given $\mathbf{s}_0$ and $\mathbf{s}_L$, extract corresponding root locations ($l_0$ and $l_L$) respectively. Here root is defined by the ground projection of hip joint. Then sample $N_R + 1$ mid locations, which forms $N_R + 2$ pairs, i.e, $[l_0, l_1], ..., [l_{N_R}, l_L]$. Distance defined by each pair $d_m$ is bounded, i.e., $\mathbf{t}_{min} < d_m < \mathbf{t}_{max}$.

- Step 2: For each location pair, we search reference subsequence for similar translation defined by $d_m$ with maximum tolerance $\sigma_t$. We conduct local motion composition for searched reference subsequence (in Section 2.2).

- Step 3: We sequentially conduct goal conditioned bi-directional motion interpolation for two consecutive reference subsequences (in Section 2.3). As a preprocessing procedure, the overall location and direction of latter sequence is adjusted w.r.t former one for interpolation. The final sequence is obtained through concatenating all synthesised clips along temporal axis.

