# OpenReview forum: "Composable Semi-parametric Modelling for Long-range Motion Generation"
_ICLR.cc/2020/Conference — Reject_

### Official Review · AnonReviewer3 · 2019-10-27
**Official Blind Review #3**

**Rating:** 3

**Review:**

This paper aims at taking techniques from motion interpolation into the regime where one is able to generate longer range motion sequences, in the domain of physically plausible computer animation of characters. In the way that the authors have set up the problem, an initial database seeds the search for plausible transitions between two given poses. So, the technique being proposed must address how to keep physical realism (the long-standing question of "dynamics filtering" along the lines of Yamane, Katsu, and Yoshihiko Nakamura. "Dynamics filter-concept and implementation of online motion generator for human figures." IEEE transactions on robotics and automation 19.3 (2003): 421-432. Of course the problem here also needs to address "style" which needs different models).

The authors propose an approach, building on recent neural network architectures including GANs and VAEs, which combines an embedding into a latent space to capture the style+content and then a Bi-LSTM model to make P-step predictions to extract the interpolating poses. The architecture is not very well explained and could be presented much better. However, the constituent elements are fairly standard ones. The learning of the embedding space is based on minimising a reconstruction loss and adapting parameters of a network that takes convolutions over time windows and stacks different channels. The LSTM training is also based on prior work in that domain.

The authors try to show with experiments that the proposed model is doing better both on diversity and accuracy in standard motion capture datasets. However, the baselines used, especially for the accuracy comparison, are more along the lines of an ablation study than a genuine comparison with alternate methods.

This is coupled with the fact that the authors do not seem to have engaged with a fairly established literature on modelling such sequences, e.g.,
Brand, M., & Hertzmann, A. (2000, July). Style machines. In Proceedings of the 27th annual conference on Computer graphics and interactive techniques (pp. 183-192). ACM Press/Addison-Wesley Publishing Co..
Li, Y., Wang, T., & Shum, H. Y. (2002, July). Motion texture: a two-level statistical model for character motion synthesis. In ACM transactions on graphics (ToG) (Vol. 21, No. 3, pp. 465-472). ACM.

These papers also solve the same problem the authors have set out to solve and arguably do pretty well. How well do they compare to the authors' approach and in what ways have they built further? It would be helpful to understand this with quantitative evidence.

As it stands, this comes across as a report on a preliminary experiment. Indeed, figure 10 is just a single learning curve without much more interpretation and analysis. The paper would be much stronger if situated better with respect to other established work and also supported by more systematic empirical experiments.

**Experience Assessment:**

I have published in this field for several years.

**Review Assessment: Checking Correctness Of Derivations And Theory:**

I carefully checked the derivations and theory.

**Review Assessment: Checking Correctness Of Experiments:**

I assessed the sensibility of the experiments.

**Review Assessment: Thoroughness In Paper Reading:**

I read the paper thoroughly.

---

> ### Author Response · Authors · 2019-11-15
> **Response to R3**
>
> Thank you for your valuable and comprehensive comments. Regarding your concerns we give response as follows:
> 1.Clarification on comparison experiments about  global composition:
>  For global composition, the most relevant work is motion prediction. However, as a motion generation task, it seems not so natural to directly compare with them. Because the motion accuracy is not a valid metric for evaluation of motion generation, which concentrates on diversity. To this end, we only compare this part with variants of our own model and merely report the training curve.
>
> 2.Discussion on related works mentioned by reviewer:
> Discussion on “Style machines”:
> This paper introduces a fully data-driven method for articulated motion generation, which needs online optimization if given new demonstration data as inputs. The involved learning procedure is relatively more complex compared to ours and needs careful hyperparameter tuning. On the contrary, our model combines both advantages of deep model and purely data-driven method.
>
> Discussion on “Motion texture”:
> This paper proposes a statistical model for approximation of original motion distribution, which is represented by a transition matrix indicating the possibility of state change. This method requires all motion data are available for generation, which is in turn restricted to seen data. Differently, our model could generalize to unseen data, which implies more practical value for downstream application.

---

### Official Review · AnonReviewer4 · 2019-10-30
**Official Blind Review #4**

**Rating:** 3

**Review:**

1. Summary:
The paper proposed ''composable semi-parametric modeling'' for generating long-range diverse and distinctive behaviors to achieve a specific goal location. The non-parametric part is a memory bank that is used to retrieve motion patterns from source materials. The parametric part contains several deep neural networks which are to compose the retrieved materials for high quality and smooth motion generation. The overall idea is novel in the sense that they aim to combine the strength of the non-parametric method (with rich pattern and diversities) and the parametric method (powerfull ability to generate coherent results). The proposed ideas are evaluated on two datasets and outperform compared approaches qualitatively.

 2. I am borderline to this paper but prone to weakly reject this paper.

   (1) My major concern about this paper is the lack of reasoning about each component.
        - local motion composition: How does the design help encode both style and content information in the outputs? Since reconstruction loss is utilized, the optimal should be to learn an identity mapping. How does the author avoid this to happen? The paper referred to the style embedding features H_s, but it is hard to understand how this is associated with the architecture. Also, the usefulness of local motion composition is not well illustrated in the experiments part.

    -global motion composition: How the network is supervised to compose different clips together? It seems that the authors again adopt reconstruction loss, how the loss penalize inconsistent predictions between clips is not well explained.

   (2) The overall presentation of this paper is hard to follow. I spent a lot of time on understanding the notations and corresponding networks. For example, the input representation is not well stated and the author referred to \phi_loc without any indication in Fig, 2. It's very hard to understand the details of the method. I would like to suggest that the author improves the writing especially for Section 2.

(3) Does the style motion shared by all sub-sequences?

(4) Figure 5 shows a sequence. however, the COSMO result seems not natural for me since the velocity in the sequence changes.

I would like to change my rating if my questions are well addressed in the rebuttal. Overall, this is an interesting idea.


**Experience Assessment:**

I do not know much about this area.

**Review Assessment: Checking Correctness Of Derivations And Theory:**

I assessed the sensibility of the derivations and theory.

**Review Assessment: Checking Correctness Of Experiments:**

I assessed the sensibility of the experiments.

**Review Assessment: Thoroughness In Paper Reading:**

I read the paper at least twice and used my best judgement in assessing the paper.

---

> ### Author Response · Authors · 2019-11-15
> **Response to R4**
>
> Thank you for your valuable and comprehensive comments. Regarding your concerns we give response as follows:
>
> 1.Clarification on local composition part:
> In our work, the motion style is defined as latent feature which keeps relatively constant throughout the sequence, while content feature changes w.r.t time stamp. Motivated by this, style and content feature is learned from completely different schemes. Style feature is extracted from the whole motion sequence and fused into one single feature, which is further concatenated to each time stamp identically for reconstruction. Differently, the content feature is modeled with only consecutive 3 time stamps in a slide window manner. The embedding features are used to construct the final style feature in the linear combination manner, where combination coefficients are generated by style encoder.
>
> 2.Clarification on local composition part:
> During training, the network is trained with the same clips but tested with different clips. We use linear interpolation at the feature space to facilitate the final motion transition between different motion clips (equ. 2.6 in our paper). We would like to emphasize that, even without GT transition of different motion clips for training, the above operation is still able to produce visually natural and long-range motion.
>
> 3.Style feature configuration for different clips:
> The style is not shared by all sub-sequences for enhancement of motion diversity.
>
> 4.Clarification on Fig.5:
> The velocity change results from motion transition. In Fig 5, the character shows a turning around the action with one foot keeping static. This is the main reason that the velocity changes.
> The video demo may be better for visualization, where the global composition part demonstrates a similar motion. Please refer to our project website.

---

### Official Review · AnonReviewer1 · 2019-10-31
**Official Blind Review #1**

**Rating:** 6

**Review:**

The paper tackles the problem of generating long-range, diverse and natural looking motion sequence between initial and end states, and proposes to use a semi-parametric approach consisting of local and global models. Specifically, first the proposed approach extracts local motion feature from a reference subsequence and style feature from another, and then generates a new motion sequence. Then, global motion composition is done to interpolated generated local subsequences by bi-directional composition. In experimental validation, the approach outperforms two baselines (GAN and VAE).

The proposed approach seems interesting and relatively novel, and technically sound. Also using disentangled representation of style and content is well-motivated. In addition, its generated motions look natural and visually appealing. However, the paper needs more thorough experimental validation to demonstrate its effectiveness better. First,  it was not quite clear which specific baselines were exploited. Are they simple VAE and GAN, or HP-GAN (Barsoum et al. (2018)) and MT-VAE (Yan et al. (2018)) ? And no animated sample of baselines was provided in the demo website. Additionally, in the global composition results, there is no other baseline except its own variants. Finally, even thought the provided figures and animated demos look appealing, it may be required to provide more qualitative results such as user study.

Writing is fine but there were many typos and it may need more proofreading. Also it might need to add more detail in the presentation.

Detailed comments:
- Why is GAN better than the proposed model with 10% of training data in Fig. 3? Do you have any explanation about it?
- The paper uses separate training of local and global models. But how about joint training of them?
- Difference between style and content is sometimes not clear, for example, rightmost one in the last sample in the demo website. Could you add more constraint or model specification to induce more disentangled representation?
- It is not quite clear if standard deviation is a good metric for motion diversity metric since visual diversity might not be directly correlated to standard deviation of joints (for example, some joints might be more important than others for inducing diversity.)

Typos:
p4: lone-range one -> long-range one
p5: (Gupta et al.) -> (Gupta et al., 2018), (Zhao et al.) -> (Zhao et al., 2019), (Wang et al.) -> (Wang et al., 2019)
p6: Our works -> Our work
p7: Fig. 3: standard diviation -> standard deviation
p8: Recall that we representation -> Recall that we do representation?
p9: As shown in Fig 7(A) -> As shown in Fig 7(B)
p10: rout constrain -> route constraint, gravity constrain -> gravity constraint


In sum, I think the paper is at the borderline but it could be improved and better by having more through experimental validation and more detailed presentation.

**Experience Assessment:**

I have published one or two papers in this area.

**Review Assessment: Checking Correctness Of Derivations And Theory:**

I carefully checked the derivations and theory.

**Review Assessment: Checking Correctness Of Experiments:**

I carefully checked the experiments.

**Review Assessment: Thoroughness In Paper Reading:**

I read the paper at least twice and used my best judgement in assessing the paper.

---

> ### Author Response · Authors · 2019-11-15
> **Response to R1**
>
> Thank you for your valuable and comprehensive comments. Regarding your concerns we give response as follows:
>
> 1.Clarification of baseline:
> Both baselines are built on original VAE and GAN, but with different configuration accordingly. For VAE, one initial state is given as a starting point, which produces the following states in a recurrent manner. KL loss is incorporated with a weight of 1e-4. For GAN, a random noise vector is given as inputs, which produces all motion states at the same time. Generator and Discriminator are both built with a 1D convolution layer for 1D temporal data.
>
> 2.Comparison over global composition:
> For global composition, the most relevant work is motion prediction. However, as a motion generation task, it seems not so natural to directly compare with them. Because the motion accuracy is not a valid metric for evaluation of motion generation, which concentrates on diversity. To this end, we only compare this part with variants of our own model and merely report the training curve.
>
> 3.Explanation on why GAN is better with 10% data:
> our work is a semi-parametric model that merely depends on the diversity of motion data and hard to reach the upper bound restricted by the temporal dependency. Differently, the input of GAN is random noise, which is probably more diverse than used 10% motion data. This lowers the learning difficulty of GAN to some extent and leads to more diverse results than ours with 10% data.
>
> 4.Motivation on disjoint training:
> We have tried joint training in our experiments. However, we observed an unstable training stage at the first few epochs which led to sub-optimal results. Meanwhile, it took more time for joint training. (multiplicative time)
>
> 5.Enhancement of disentanglement learning:
> This is exactly our future work, i.e.,  to pursue more explicit disentanglement learning. One direction looks promising is bilinear transformation for explicit content/style disentangled representation. More specifically, through applying content/style transformation with row/column multiplication respectively, the influence of content / style feature is explicitly decomposed.
>
> 6. We will carefully revise our paper for typos.

---

### Decision · Program_Chairs · 2019-12-19

**Decision:**

Reject

**Comment:**

The submission presents a semi-parametric approach to motion synthesis. The reviewers expressed concerns about the presentation, the relationship to existing work, and the scope of the results. After the authors' responses and revision, concerns remain. The AC also notes that the submission is 10 pages long. The AC recommends rejecting the submission.